# Particle Filtering for Localization of Broadband Sound Source Using an Ocean-Bottom Seismometer Sensor

**DOI:** 10.3390/s19102236

**Published:** 2019-05-14

**Authors:** Yaqin Liu, Haigang Zhang, Ziyang Li, Xiaohan Wang, Jun Ma

**Affiliations:** 1Acoustic Science and Technology Laboratory, Harbin Engineering University, Harbin 150001, China; liuyaqin@hrbeu.edu.cn (Y.L.); 397976970@hrbeu.edu.cn (Z.L.); wangxiaohan@hrbeu.edu.cn (X.W.); majun@hrbeu.edu.cn (J.M.); 2Key Laboratory of Marine Information Acquisition and Security, Harbin Engineering University, Ministry of Industry and Information Technology, Harbin 150001, China; 3College of Underwater Acoustic Engineering, Harbin Engineering University, Harbin 150001, China

**Keywords:** OBS sensor, particle filtering, vertical wave impedance

## Abstract

Passive source localization is a challenging task for one receiver, and the pressure sensor provides relatively simple information. An ocean-bottom seismometer (OBS) sensor placed on the seafloor surface can provide more information—not only pressure information, but also three-axis (*x*-, *y*-, and *z*-axis) velocity information at the seafloor interface. In this paper, an OBS sensor was used to estimate the position of the broadband sound source in a Pekeris shallow water waveguide with elastic bottom. As the dynamics that characterize ocean acoustic applications are inherently nonlinear, non-Gaussian, and non-stationary processes that quickly vary with space and time, sequential Bayesian filtering, such as particle filtering (PF), is able to adapt to these environmental changes. Simulation results show that the PF method with the vertical wave impedance (the ratio of the pressure and vertical particle velocity) in the frequency domain as a measurement vector is not affected by source depth and source spectrum information, making it more tolerant and more robust than that with pressure in positioning. Experimental data results verified the effectiveness of the PF method with the vertical wave impedance for the localization of the explosive source.

## 1. Introduction

Localization of an underwater sound source is an important practical problem in underwater acoustics. Of all the methods for source localization, matched-field processing (MFP) attracted a huge amount of interest over the years. MFP is a technique combining hydro-acoustic physics and signal processing technology that made important progress in underwater acoustic positioning. However, the matched field method uses a deterministic model, which is plagued by mismatch problems, including environmental mismatch, statistical mismatch, and system mismatch [1]. Sequential filtering provides a suitable framework for estimating and updating unknown parameters of a system as data become available. Moreover, sequential filtering is demonstrated to be a powerful estimation tool, employing prediction from previous estimates and updates stemming from physical and statistical models that relate acoustic measurements to the unknown parameters [2]. An adaptive model-based approach using the state-space formulation was for the first time implemented by Candy et al. [3]. Coupling a nonlinear optimization algorithm with the extended Kalman filter (EKF)-based ocean acoustic model can solve the source localization problem in a complex ocean environment. This approach is capable of solving the mismatch problem to some extent. Furthermore, Candy et al. [4] provided a model-based Bayesian processor to estimate the bearings of moving sources using horizontally towed array data. The processor uses the particle filtering (PF) as the estimator, which has better tracking performance than classical nonlinear filters (e.g., extended/unscented Kalman filters). A particle filtering method for the estimation of such arrivals was presented by Michalopoulou and Jain [5,6]. Furthermore, successful localization with real data was demonstrated using arrival times and corresponding probability density functions (PDFs) extracted via particle filtering [7]. These researches generally used sound pressure signals received by the hydrophone array. In Reference [8], the vertical specific acoustic impedance (the ratio of the complex pressure field and vertical pressure gradient) was used to sense ocean-bottom geo-acoustic properties in the Pekeris environment using a sequential approach. This approach was applied to data collected off the Senegalese coast. Previous studies were generally based on the fluid seabed, without considering the effects of shear waves. However, the seabed in the real marine environment is generally an elastic medium. Therefore, we derive the vertical wave impedance which the ocean-bottom seismometer (OBS) sensor obtained in Pekeris waveguide with an elastic bottom. Then, a passive ranging method for the broadband acoustic source using vertical wave impedance is proposed in this paper based on the particle filtering approach.

The remainder of this paper is organized as follows: in Section 2, the pressure and vertical wave impedance in the frequency domain are constructed in Pekeris waveguide with elastic bottom. In Section 3, the PF framework for estimating the position of the source is introduced. In Section 4, the positioning performances of the pressure and vertical wave impedance as the measurement vector are simulated and analyzed based on the PF method. In Section 5, the presented estimation localization method is employed to process the experimental data. Conclusions are given in Section 6.

## 2. Theoretical Modeling

In the cylindrical coordinate system, the Pekeris waveguide with elastic bottom, as shown in Figure 1, was built to obtain the pressure and vertical wave impedance received by the OBS sensor. In this section, the pressure field and velocity field expressions are established in normal mode based on wavenumber integration approaches. Thus, in Section 2.1, we briefly introduce the derivation process of potential functions based on wavenumber integration approaches; then, in Section 2.2, the potential function expressions are represented in normal mode form, and then the pressure and the vertical wave impedance at the seafloor interface are obtained.

### 2.1. Potential Functions

The depth of the uniform water layer is denoted as *H*, and the density and the sound speed of water are *ρ*_1_ and *c*_1_, respectively. The elastic bottom is assumed to be homogeneous and semi-infinite. The density, the compressional wave, and the shear wave speed of the bottom are constants that are expressed as *ρ*_2_, *c_p_*, and *c_s_*, respectively. As shown in Figure 1, the horizontal axis is range *r*, the vertical axis is depth *z*, the plane *z* = 0 is considered as the sea surface, and the broadband source is positioned at (0, *z_s_*). The spectrum of the source is denoted as *S*(*ω*). The OBS sensor is placed on the seafloor interface as a receiver. The potential function of sound field in water is denoted as *φ*_1_, and potential function of sound field in elastic medium is the sum of the scalar potential function *φ*_2_ and vector potential function $\vec{\psi }$. In the cylindrically symmetric case, only the component along the *θ* direction exists for $\vec{\psi }$, which is denoted as *ψ* referring to Reference [9]. The derivation process of potential functions is detailed in Appendix A.

The expression of $\varphi_{1},\varphi_{2},\psi$ can be written as
(1)$$\varphi_{1}\left(r,z\right)=\{\begin{array}{c} \int_{0}^{\infty }\frac{2\sin\beta_{1}z}{\beta_{1}}\left[\frac{\beta_{1}\cos\beta_{1}\left(H-z_{s}\right)-ib\beta_{2}K\sin\beta_{1}\left(H-z_{s}\right)}{\beta_{1}\cos\beta_{1}H-ib\beta_{2}K\sin\beta_{1}H}\right]J_{0}\left(\xi r\right)\xi d\xi ,0\leq z<z_{s}\\ \int_{0}^{\infty }\frac{2\sin\beta_{1}z_{s}}{\beta_{1}}\left[\frac{\beta_{1}\cos\beta_{1}\left(H-z\right)-ib\beta_{2}K\sin\beta_{1}\left(H-z\right)}{\beta_{1}\cos\beta_{1}H-ib\beta_{2}K\sin\beta_{1}H}\right]J_{0}\left(\xi r\right)\xi d\xi ,z_{s}\leq z<H \end{array},$$
(2)$$\varphi_{2}=-\int_{0}^{\infty }\frac{2\xi \sigma K}{\chi^{2}}\frac{2b\sin\beta_{1}z_{s}e^{i\beta_{2}\left(z-H\right)}}{\beta_{1}\cos\beta_{1}H-ib\beta_{2}K\sin\beta_{1}H}J_{0}\left(\xi r\right)\xi d\xi \begin{array}{cc} & z>H \end{array},$$
(3)$$\psi =\int_{0}^{\infty }\frac{2i\xi \beta_{2}K}{\chi^{2}}\frac{2b\sin\beta_{1}z_{s}e^{i\gamma \left(z-H\right)}}{\beta_{1}\cos\beta_{1}H-ib\beta_{2}K\sin\beta_{1}H}J_{1}\left(\xi r\right)\xi d\xi \begin{array}{cc} & z>H \end{array},$$where $\beta_{1}=\sqrt{k_{1}^{2}-\xi^{2}}$, $\beta_{2}=\sqrt{k_{2}^{2}-\xi^{2}}$, $\gamma =\sqrt{\chi^{2}-\xi^{2}}$, $k_{1}=\omega /c_{1}$, $k_{2}=\omega /c_{p}$, $\chi =\omega /c_{s}$, $b=\rho_{1}/\rho_{2}$, $\sigma =\left(2\xi^{2}-\chi^{2}\right)/2\xi$,$K=\chi^{4}/\left[4\xi^{2}\left(\sigma^{2}+\beta_{2}\gamma \right)\right]$, *ξ* is horizontal wavenumber, and *ω* is the angular frequency.

### 2.2. Pressure and Vertical Wave Impedance

According to the formula
$$\begin{array}{c} J_{0}\left(\xi r\right)=\frac{1}{2}\left[H_{0}^{\left(1\right)}\left(\xi r\right)+H_{0}^{\left(2\right)}\left(\xi r\right)\right]\\ H_{0}^{\left(2\right)}\left(\xi r\right)=-H_{0}^{\left(1\right)}\left(\xi re^{i\pi }\right) \end{array},$$the first integral in Equation (1) can be rewritten as
(4)$$\varphi_{1}\left(r,z\right)=\int_{-\infty }^{\infty }\frac{\sin\beta_{1}z}{\beta_{1}}\left[\frac{\beta_{1}\cos\beta_{1}\left(H-z_{s}\right)-ib\beta_{2}K\sin\beta_{1}\left(H-z_{s}\right)}{\beta_{1}\cos\beta_{1}H-ib\beta_{2}K\sin\beta_{1}H}\right]H_{0}^{\left(1\right)}\left(\xi r\right)\xi d\xi ,0\leq z<z_{s},$$with $H_{0}^{\left(1\right)}\left(\cdot \right),$ as the first kind of zero-order Hankel function.

By Cauchy’s method of residues, the integral of Equation (4) will be equal to
(5)$$\begin{array}{ll} \varphi_{1}\left(r,z\right) & =\mathrm{Res}\left\{Z_{1}\left(z,\xi \right)H_{0}^{\left(1\right)}\left(\xi r\right)\xi \right\}-\int_{branch\begin{array}{c} line \end{array}}Z_{1}\left(z,\xi \right)H_{0}^{\left(1\right)}\left(\xi r\right)\xi d\xi \\ & =\varphi_{N}\left(r,z\right)+\varphi_{L}\left(r,z\right) \end{array},$$where $\varphi_{N}\left(r,z\right),$ is called the normal mode, and $\varphi_{L}\left(r,z\right)$ is called the lateral wave. 

The expression of $\varphi_{N}\left(r,z\right)$ is written as
(6)$$\begin{array}{ll} \varphi_{N}\left(r,z\right) & =2\pi i\sum_{n}\frac{\sin\beta_{1n}z}{\beta_{1n}}\frac{\beta_{1n}\cos\beta_{1n}\left(H-z_{s}\right)-ib\beta_{2n}K\sin\beta_{1n}\left(H-z_{s}\right)}{\frac{\partial }{\partial \xi_{n}}\left(\beta_{1n}\cos\beta_{1n}H-ib\beta_{2n}K\sin\beta_{1n}H\right)}H_{0}^{\left(1\right)}\left(\xi_{n}r\right)\xi_{n}\\ & =2\pi i\sum_{n}F_{n}^{2}\sin\beta_{1n}z\sin\beta_{1n}z_{s}H_{0}^{\left(1\right)}\left(\xi_{n}r\right) \end{array},$$where
$$F_{n}^{2}=\frac{\beta_{1n}}{H\beta_{1n}-\sin\beta_{1n}H\cos\beta_{1n}H-b^{2}K_{n}^{2}\tan\beta_{1n}H\sin^{2}\beta_{1n}H-ib\beta_{1n}\beta_{2n}\frac{\partial K_{n}}{\partial \xi_{n}}\sin^{2}\beta_{1n}H/\xi_{n}}$$
$$\frac{\partial K_{n}}{\partial \xi_{n}}=-\frac{\chi^{4}}{4}\frac{2\xi_{n}\left(\sigma_{n}^{2}+\beta_{2n}\gamma_{n}\right)+\xi_{n}^{2}\left[2\sigma_{n}\left(1+\frac{\chi^{2}}{2\xi_{n}^{2}}\right)-\beta_{2n}\frac{\xi_{n}}{\gamma_{n}}-\frac{\xi_{n}}{\beta_{2n}}\gamma_{n}\right]}{\xi_{n}{}^{4}{\left(\sigma_{n}{}^{2}+\beta_{2n}\gamma_{n}\right)}^{2}}$$and values of $\xi_{n}$ are determined by the dispersion equation, i.e., $\beta_{1n}\cos\beta_{1n}H-ib\beta_{2n}K_{n}\sin\beta_{1n}H=0$.

Similarly, $\varphi_{N}\left(r,z\right)$ in the depth domain $z_{s}\leq z<H$ also can be obtained with the same expression as Equation (6).

Due to the effect of the shear wave, the expression of the lateral wave is more complicated than that in Reference [9]. Specifically, the lateral wave mainly consists of two parts. One integration along the branch at first runs from $k_{2}+i\infty$ to $k_{2}$ along the left side of the branch line with the negative imaginary part of $\beta_{2}$, then runs from $k_{2}$ to $k_{2}+i\infty$ along the right side of the branch line with the positive imaginary part of $\beta_{2}$. The other integration along the branch at first runs from χ+i∞ to χ along the left side of the branch line with the negative imaginary part of γ, then runs from χ to χ+i∞ along the right side of the branch line with the positive imaginary part of γ. Under the assumption that $k_{2}<\chi <k_{1}$, the approximate expression of lateral wave is
(7)$$\varphi_{L}\left(r,z\right)=\varphi_{Lk}\left(r,z\right)+\varphi_{L\chi }\left(r,z\right),$$where
$$\begin{array}{ll} \varphi_{Lk}\left(r,z\right) & =\int_{k_{2}}^{k_{2}+i\infty }\frac{\sin\beta_{1}z}{\beta_{1}}\{\frac{{\left(2\xi^{2}-\chi^{2}\right)}^{2}\beta_{1}\cos\beta_{1}\left(H-z_{s}\right)+\beta_{2}\left[4\xi^{2}\gamma \beta_{1}\cos\beta_{1}\left(H-z_{s}\right)-ib\chi^{4}\sin\beta_{1}\left(H-z_{s}\right)\right]}{{\left(2\xi^{2}-\chi^{2}\right)}^{2}\beta_{1}\cos\beta_{1}H+\beta_{2}\left[4\xi^{2}\gamma \beta_{1}\cos\beta_{1}H-ib\chi^{4}\sin\beta_{1}H\right]}\\ & -\frac{{\left(2\xi^{2}-\chi^{2}\right)}^{2}\beta_{1}\cos\beta_{1}\left(H-z_{s}\right)-\beta_{2}\left[4\xi^{2}\gamma \beta_{1}\cos\beta_{1}\left(H-z_{s}\right)-ib\chi^{4}\sin\beta_{1}\left(H-z_{s}\right)\right]}{{\left(2\xi^{2}-\chi^{2}\right)}^{2}\beta_{1}\cos\beta_{1}H-\beta_{2}\left[4\xi^{2}\gamma \beta_{1}\cos\beta_{1}H-ib\chi^{4}\sin\beta_{1}H\right]}\}H_{0}^{\left(1\right)}\left(\xi r\right)\xi d\xi \end{array}$$
(8)$$\approx \{\begin{array}{c} \frac{2bk_{2}\chi^{4}\sin\sqrt{k_{1}^{2}-k_{2}^{2}}z_{0}\sin\sqrt{k_{1}^{2}-k_{2}^{2}}z}{\left(k_{1}^{2}-k_{2}^{2}\right)\cos^{2}\sqrt{k_{1}^{2}-k_{2}^{2}}H}\frac{1}{r^{2}}e^{i\left(k_{2}r-\frac{\pi }{2}\right)},if\cos\sqrt{k_{1}^{2}-k_{2}^{2}}H\neq 0\\ -\frac{2b{\left(2k_{2}{}^{2}-\chi^{2}\right)}^{2}\chi^{4}\sin\sqrt{k_{1}^{2}-k_{2}^{2}}z_{0}\sin\sqrt{k_{1}^{2}-k_{2}^{2}}z}{\left[4k_{2}^{2}\gamma \sqrt{k_{1}^{2}-k_{2}^{2}}\cos\sqrt{k_{1}^{2}-k_{2}^{2}}H-ib\chi^{4}\sin\sqrt{k_{1}^{2}-k_{2}^{2}}H\right]}\frac{1}{r}e^{ik_{2}r},if\cos\sqrt{k_{1}^{2}-k_{2}^{2}}H=0 \end{array},$$
(9)$$\begin{array}{l} \varphi_{L\chi }\left(r,z\right)=\int_{\chi }^{\chi +i\infty }\frac{\sin\beta_{1}z}{\beta_{1}}\{\frac{{\left(2\xi^{2}-\chi^{2}\right)}^{2}\beta_{1}\cos\beta_{1}\left(H-z_{s}\right)-ib\beta_{2}\chi^{4}\sin\beta_{1}\left(H-z_{s}\right)+4\xi^{2}\beta_{2}\gamma \beta_{1}\cos\beta_{1}\left(H-z_{s}\right)}{{\left(2\xi^{2}-\chi^{2}\right)}^{2}\beta_{1}\cos\beta_{1}H-ib\beta_{2}\chi^{4}\sin\beta_{1}H+4\xi^{2}\beta_{2}\gamma \beta_{1}\cos\beta_{1}H}\\ \;\;\;\;-\frac{{\left(2\xi^{2}-\chi^{2}\right)}^{2}\beta_{1}\cos\beta_{1}\left(H-z_{s}\right)-ib\beta_{2}\chi^{4}\sin\beta_{1}\left(H-z_{s}\right)-4\xi^{2}\beta_{2}\gamma \beta_{1}\cos\beta_{1}\left(H-z_{s}\right)}{{\left(2\xi^{2}-\chi^{2}\right)}^{2}\beta_{1}\cos\beta_{1}H-ib\beta_{2}\chi^{4}\sin\beta_{1}H-4\xi^{2}\beta_{2}\gamma \beta_{1}\cos\beta_{1}H}\}H_{0}^{\left(1\right)}\left(\xi r\right)\xi d\xi \\ \approx \{\begin{array}{c} \frac{8\sin\sqrt{k_{1}^{2}-\chi^{2}}z_{s}\sin\sqrt{k_{1}^{2}-\chi^{2}}z}{b\chi \sin^{2}\sqrt{k_{1}^{2}-\chi^{2}}H}\frac{1}{r^{2}}e^{i\left(\chi r+\frac{\pi }{2}\right)},if\cos\sqrt{k_{1}^{2}-\chi^{2}}H=0\\ -\frac{\chi^{2}b\sin\sqrt{k_{1}^{2}-\chi^{2}}z_{s}\sin\sqrt{k_{1}^{2}-\chi^{2}}z}{2\left(k_{1}^{2}-\chi^{2}\right)\cos^{2}\sqrt{k_{1}^{2}-\chi^{2}}H}\frac{1}{r}e^{i\chi r},if\left(\begin{array}{c} \cos\sqrt{k_{1}^{2}-\chi^{2}}H\neq 0,\\ b\sqrt{\chi^{2}-k_{2}^{2}}\tan\sqrt{k_{1}^{2}-\chi^{2}}H+\sqrt{k_{1}^{2}-\chi^{2}}=0 \end{array}\right)\\ \frac{8\left(k_{2}^{2}-\chi^{2}\right)b\sin\sqrt{k_{1}^{2}-\chi^{2}}z_{s}\sin\sqrt{k_{1}^{2}-\chi^{2}}z}{\chi {\left[\sqrt{k_{1}^{2}-\chi^{2}}\cos\sqrt{k_{1}^{2}-\chi^{2}}H+b\sqrt{\chi^{2}-k_{2}^{2}}\sin\sqrt{k_{1}^{2}-\chi^{2}}H\right]}^{2}}\frac{1}{r^{2}}e^{i\left(\chi r-\frac{\pi }{2}\right)},if\left(\begin{array}{c} \cos\sqrt{k_{1}^{2}-\chi^{2}}H\neq 0,\\ b\sqrt{\chi^{2}-k_{2}^{2}}\tan\sqrt{k_{1}^{2}-\chi^{2}}H+\sqrt{k_{1}^{2}-\chi^{2}}\neq 0 \end{array}\right) \end{array} \end{array}$$

Equations (8) and (9) show that, in general, lateral waves attenuate more quickly than spherical waves during propagation, and will possess a significant value only for distances not far from the source. In order to reflect the phenomenon more intuitively, the transmission losses (TLs) of normal modes, the lateral wave, and the spherical wave are shown in Figure 2. The simulation environment parameters are shown in Table 1, the sound source depth is 20 m, and the receiver is placed on the seafloor interface. Here, the frequency is 50 Hz. Figure 2 shows that the lateral wave attenuation decays faster with distance. Thus, in general, when long-range sound transmission is considered, the contribution of the lateral wave usually can be neglected and
(10)$$\varphi_{1}\left(r,z\right)\approx \varphi_{N}\left(r,z\right)=2\pi i\sum_{n}F_{n}^{2}\sin\beta_{1n}z\sin\beta_{1n}z_{s}H_{0}^{\left(1\right)}\left(\xi_{n}r\right).$$

In the fluid, the relationships between potential function, complex pressure, and the vertical particle velocity at angular frequency *ω* are as follows:(11)$$p\left(r,z,\omega \right)=\rho_{1}\omega^{2}S\left(\omega \right)\varphi_{1}\left(r,z\right)=2\pi iS\left(\omega \right)\rho_{1}\omega^{2}\sum_{n}F_{n}^{2}\sin\beta_{1n}z\sin\beta_{1n}z_{s}H_{0}^{\left(1\right)}\left(\xi_{n}r\right);$$
(12)$$v_{z}\left(r,z,\omega \right)=-i\omega S\left(\omega \right)\frac{\partial \varphi_{1}\left(r,z\right)}{\partial z}=\omega 2\pi S\left(\omega \right)\sum_{n}F_{n}^{2}\beta_{1n}\cos\beta_{1n}z\sin\beta_{1n}z_{s}H_{0}^{\left(1\right)}\left(\xi_{n}r\right)$$

Because the vertical particle velocity is continuous at *z* = *H*, the ratio of the pressure and the vertical velocity received by the OBS is
(13)$$Z_{z}\left(r,H,\omega \right)=\frac{p\left(r,H,\omega \right)}{v_{z}\left(r,H,\omega \right)}=i\rho_{1}\omega \frac{\sum_{n}F_{n}^{2}\sin\beta_{1n}H\sin\beta_{1n}z_{s}H_{0}^{\left(1\right)}\left(\xi_{n}r\right)}{\sum_{n}F_{n}^{2}\beta_{1n}\cos\beta_{1n}H\sin\beta_{1n}z_{s}H_{0}^{\left(1\right)}\left(\xi_{n}r\right)}.$$

If only the single normal mode is considered, the expression of vertical wave impedance is represented as (*Z_z_* = *i**ρ*_1_*ω*sin*β*_1*n*_*H*/*β*_1*n*_cos*β*_1*n*_*H*). Hence, the vertical wave impedance is only related to the receiver depth and is independent of the distance. Therefore, for ranging, the vertical wave impedance must take into account the normal mode order of two and above.

## 3. PF Framework

PF is a sequential Monte Carlo (MC) method employing the sequential estimation of relevant probability distributions using the importance sampling (IS) techniques and the approximations of distributions with discrete random measures [10,11,12,13]. PF is a technique to implement sequential Bayesian estimators via MC simulation [14]. PF has the advantage that the noise model and the system model are not limited. Compared with the Kalman filter, PF can be applied to nonlinear systems and is not limited by Gaussian distribution. The PF framework for estimating the position of the source is organized as follows: in Section 3.1, a general background about the state-space equation for the estimation of evolving parameters in dynamical systems is given together with the basics of Bayesian filtering. In Section 3.2, the filter equations are derived starting from the basic IS concepts, moving to sequential importance sampling (SIS), finally deriving the commonly used PF, often referred to as sequential importance resampling (SIR).

### 3.1. Background

#### 3.1.1. State-Space Model

The state-space model requires two equations: the state equation and the measurement equation. The state equation models the evolution of the state vector over time. The measurement equation performs the mapping from state vectors to observation vectors. The two equations are represented as follows:(14)$$x_{k}=f\left(x_{k-1}\right)+w_{k-1},$$
(15)$$y_{k}=d\left(x_{k}\right)+v_{k},$$where **x** = [*r*, *z*_s_]^T^, *r* is the range, *z*_s_ is the source depth, and T represents the transpose of matrix **x**. 

The state Equation (14) describes the evolution or transition of **x_k_** and assumes that states follow a first-order Markov process. Function **f**(∙) is the state prediction operator, which models the evolution of the state vector at step *k* − 1 to that of step *k*. In this work, **f**(∙) is taken as the unit matrix **I**. The measurement Equation (15) relates measurements y_k_ to state vector x_k_ through function **d**(∙), where **y** is the measurement vector. Function **d**(∙) is the nonlinear function that relates the environmental and source parameters **x**_k_ to the acoustic measurement vector **y***_k_*. In this work, when vertical wave impedance selected as measurement vector, **y** = [*Z_z_*(**x**, *ω*_1_), …, *Z_z_* (**x**, *ω_m_*), …, *Z_z_* (**x**, *ω*_M_)]^T^, and *Z_z_* (**x**, *ω_m_*) is the vertical wave impedance at angular frequency *ω_m_*, with m∈[1,M]. Furthermore, **d**(**x**) = [d_1_(**x**), …, d*_m_*(**x**), …, d_M_(**x**)]^T^, with
(16)$$d_{m}\left(x\right)=i\rho_{1}\omega_{m}\frac{\sum_{n}F_{nm}^{2}\sin\beta_{1nm}H\sin\beta_{1nm}z_{s}H_{0}^{\left(1\right)}\left(\xi_{nm}r\right)}{\sum_{n}F_{nm}^{2}\beta_{1nm}\cos\beta_{1nm}H\sin\beta_{1nm}z_{s}H_{0}^{\left(1\right)}\left(\xi_{nm}r\right)}.$$Moreover, **w***_k_* and **v***_k_* are the state noise vector and the measurement noise vector, respectively, with
(17)$$E\left\{w_{k}w_{i}^{T}\right\}=Q_{k}\delta_{ki},E\left\{v_{k}v_{i}^{T}\right\}=R_{k}\delta_{ki},E\left\{w_{k}v_{i}^{T}\right\}=0,\forall i,k,$$where *δ*(∙) is the Dirac delta function, and **Q***_k_* and **R***_k_* are the covariance matrices at *k* for the corresponding noise terms. The modal/range uncertainty can be lumped as a state process noise term to represent sound-speed profile errors, errors in the boundary conditions, sea state effects, and ocean inhomogeneities, whereas the measurement noise can be lumped into an additive noise term to represent the near-field acoustic noise field, flow noise on the sensors, and electronic noise [15].

#### 3.1.2. Bayesian Filtering

After giving the state equation and the measurement equation, we briefly discuss the algorithm for our estimation problem. The basic problem we pursue in this paper can now be defined in terms of our mathematical models as follows:

GIVEN a set of noisy vertical wave impedances along with the state equation and measurement equation (Equations (14) and (15)) with unknown parameters (*r*, *z*_s_), FIND the “best” estimated values. 

Examining the problem from a Bayesian standpoint [16,17], we are interested in deriving the full posterior PDF for **x***_k_*. The initial PDF of the state vector, Pr(**x_0_**), is assumed to be known. Let **D***_k_* = [**y**_1_*,*
**y**_2_*, …,*
**y***_k_*] be the set of data from 1 to *k* steps. The aim is to estimate Pr(x*_k_*|D*_k_*), the posterior PDF of the state vector at step *k*. 

With the posterior PDF Pr(**x***_k_*
_− 1_|**D***_k_*
_− 1_) available, we can predict Pr(**x***_k_*|D*_k_*
_− 1_) through the transition PDF Pr(**x***_k_*|**x***_k_*
_− 1_). Due to the first-order Markov chain assumption of **x***_k_*, Pr(**x***_k_*|**x***_k_*
_− 1_) does not depend on data **D***_k_*
_− 1_. Density Pr(**x***_k_*|**D***_k_*
_− 1_) can be written as
(18)$$\begin{array}{ll} \mathrm{Pr}\left(x_{k}|D_{k-1}\right) & =\int \mathrm{Pr}\left(x_{k}|x_{k-1},D_{k-1}\right)\times \mathrm{Pr}\left(x_{k-1}|D_{k-1}\right)dx_{k-1}\\ & =\int \mathrm{Pr}\left(x_{k}|x_{k-1}\right)\times \mathrm{Pr}\left(x_{k-1}|D_{k-1}\right)dx_{k-1} \end{array}$$

When a new measurement **y***_k_* becomes available, the posterior PDF Pr(**x***_k_*|**D***_k_*) can be calculated by the Bayes theorem.
(19)$$\begin{array}{ll} \mathrm{Pr}\left(x_{k}|D_{k}\right) & =\frac{\mathrm{Pr}\left(y_{k}|x_{k}\right)\mathrm{Pr}\left(x_{k-1}|D_{k-1}\right)}{\mathrm{Pr}\left(y_{k}|D_{k-1}\right)}\\ & =\frac{\mathrm{Pr}\left(y_{k}|x_{k}\right)\mathrm{Pr}\left(x_{k-1}|D_{k-1}\right)}{\int \mathrm{Pr}\left(y_{k}|x_{k}\right)\mathrm{Pr}\left(x_{k-1}|D_{k-1}\right)dx_{k}} \end{array}$$

The posterior PDF Pr(**x***_k_*|**D***_k_*) contains all information provided from the data, the measurement equation, and the noise model about target **x***_k_* at step *k*.

### 3.2. Particle Filter

PFs track the posterior PDF Pr(**x***_k_*|**D***_k_*) using a cloud of particles ${\left\{x_{k}^{i}\right\}}_{i=1}^{N_{p}}=\left\{x_{k}^{1},x_{k}^{2},\cdots ,x_{k}^{N_{p}}\right\}$ that evolve with step *k*. Before presenting the details of the PF, we summarize the basic IS concepts.

#### 3.2.1. Importance Sampling

For a general nonlinear system, it is difficult to obtain an analytical solution of the posterior probability, and it is difficult to obtain the integral in Equations (18) and (19). In order to solve the integral problem, we introduce the MC method.

Assume that we want to compute an integral *I* = ∫*f*(*x*)d*x*. One way of computing *I* is assuming *x* is a random variable, defining *f*(*x*) = *g*(*x*)*p*(*x*), and rewriting it in the form of an expectation [2].
(20)I=E{g(x)}=∫g(x)p(x)dx,where *g*(*x*) is some function of *x* with PDF *p*(*x*). By drawing *N*_p_ independent and identically distributed *x* samples from *p*(*x*), *I* can be computed numerically via MC integration [2].
(21)$${\left\{x^{i}\right\}}_{i=1}^{N_{p}}∼p\left(x\right)\to I\approx \frac{1}{N_{p}}\sum_{i=1}^{N_{p}}g\left(x^{i}\right).$$

However, in many cases, it is too costly or not possible to sample from *p*(*x*). To mitigate difficulties with inability to directly from a posterior distribution, IS is introduced. IS is a method to compute expectations with respect to one density using random samples drawn from another. Using a simple function *q*(*x*) as the sampling density, Equation (20) can be rewritten as
(22)$$\begin{array}{l} I=\int \left[g\left(x\right)\frac{p\left(x\right)}{q\left(x\right)}\right]q\left(x\right)dx=E\left\{g\left(x\right)\frac{p\left(x\right)}{q\left(x\right)}\right\}\\ {\left\{x^{i}\right\}}_{i=1}^{N_{p}}∼q\left(x\right) \end{array}$$

The estimate is obtained using MC integration,
(23)$$\hat{I}=\frac{1}{N_{p}}\sum_{i=1}^{N_{p}}g\left(x^{i}\right)\frac{p\left(x^{i}\right)}{q\left(x^{i}\right)}=\frac{1}{N_{p}}\sum_{i=1}^{N_{p}}w^{i}g\left(x^{i}\right),$$where $w^{i}=p\left(x^{i}\right)/q\left(x^{i}\right)$ represents the importance weights.

#### 3.2.2. Sequential Importance Sampling

Bayesian filtering requires performing successive IS runs at each *k*. The output of each IS run is used as the prior for the next one. This process is referred to as SIS. Let X_k_ = [**x**_1_*,*
**x**_2_*, …,*
**x***_k_*]; it is possible to obtain posterior PDF Pr(**x***_k_*|**D***_k_*) from the full posterior density Pr(**X***_k_*|**D***_k_*).
(24)$$\mathrm{Pr}\left(x_{k}|D_{k}\right)=\int \delta \left(x_{k}-x_{k}{}^{\prime }\right)\mathrm{Pr}\left(X_{k}{}^{\prime }|D_{k}\right)dX_{k}{}^{\prime }.$$

Selecting a sampling density *q*(**X***_k_*|**D***_k_*) and implementing IS, we can obtain
(25)$$\mathrm{Pr}\left(x_{k}|D_{k}\right)\approx \sum_{i=1}^{N_{p}}W_{k}^{i}\delta \left(x_{k}-x_{k}^{i}\right),$$
(26)$$W_{k}^{i}\propto \frac{\mathrm{Pr}\left(X_{k}^{i}|D_{k}^{i}\right)}{q\left(X_{k}^{i}|D_{k}^{i}\right)}.$$

Expanding the full posterior PDF [10], Pr(**X***_k_*|**D***_k_*) can be expressed as
(27)$$\mathrm{Pr}\left(X_{k}|D_{k}\right)=\frac{\mathrm{Pr}\left(y_{k}|x_{k}\right)\mathrm{Pr}\left(x_{k}|x_{k-1}\right)}{\mathrm{Pr}\left(y_{k}|D_{k-1}\right)}\mathrm{Pr}\left(X_{k-1}|D_{k-1}\right).$$

Selecting *q*(**X***_k_*|**D***_k_*) as
(28)$$q\left(X_{k}|D_{k}\right)=q\left(x_{k}|x_{k-1},D_{k}\right)q\left(X_{k-1}|D_{k-1}\right),$$the weight of the *i*th particle at step *k* can be represented as
(29)$$\begin{array}{ll} W_{k}^{i} & \propto \frac{\mathrm{Pr}\left(y_{k}|x_{k}^{i}\right)\mathrm{Pr}\left(x_{k}^{i}|x_{k-1}^{i}\right)}{q\left(x_{k}^{i}|x_{k-1}^{i},D_{k}\right)}\times \frac{\mathrm{Pr}\left(X_{k-1}|D_{k-1}\right)}{q\left(X_{k-1}^{i}|D_{k-1}^{i}\right)}\\ & \propto \frac{\mathrm{Pr}\left(y_{k}|x_{k}^{i}\right)\mathrm{Pr}\left(x_{k}^{i}|x_{k-1}^{i}\right)}{q\left(x_{k}^{i}|x_{k-1}^{i},D_{k}\right)}W_{k-1}^{i} \end{array}$$

There are a variety of the PF algorithms available, each evolving from a particular choice of sampling density; however, perhaps the simplest is the bootstrap technique [18], which we apply to our problem. Here, the sampling density is selected as
(30)$$q\left(x_{k}^{}|x_{k-1}^{},D_{k}\right)=\mathrm{Pr}\left(x_{k}^{}|x_{k-1}^{}\right).$$

Thus, Equation (29) is simplified to
(31)$$W_{k}^{i}=\mathrm{Pr}\left(y_{k}|x_{k}^{i}\right)W_{k-1}^{i}.$$

Now, only Pr(**y***_k_*|**x***_k_*) at step *k* is employed in updating weights $W_{k}^{i}$. Pr(**y***_k_*|**x***_k_*) is called the likelihood function.

#### 3.2.3. Sequential Importance Resampling

One of the major problems with SIS is the degeneracy of the particles. After a few iterations of successive SIS, the process leads to a cloud containing few particles with large weights and numerous particles with negligible weights. This loss of sample diversity results in poor filter performance. Thus, there is a need to somehow resolve this degeneracy problem. This requirement leads to the idea of “resampling” the particles. SIS with an additional resampling step to avoid degeneracy called SIR [16,17,19], and SIR is the most popular PF implementation.

Resampling is easily performed at the end of each step *k*; alternatively, resampling is implemented when the effective number of particles *N*_eff_ needed to maintain diversity drops below a threshold *N*_thresh_. An estimate of the effective number of particles is given by
(32)$$N_{eff}=\frac{1}{\sum_{i=1}^{N_{p}}{\left(W_{k}^{i}\right)}^{2}}.$$

When *N*_eff_ is less than the threshold, resampling is performed.
(33)Neff={≤NthreshResample>NthreshAccept.

In summary, the SIR particle filter works as follows: suppose that, at time *k* – 1, there is a particle cloud $\left\{x_{k-1}^{1},x_{k-1}^{2},\cdots ,x_{k-1}^{N_{p}}\right\}$ of size *N*_p_ that, with associated weights, samples from the posterior PDF Pr(x_k − 1_|D_k − 1_). Then, transform cloud $\left\{x_{k-1}^{1},x_{k-1}^{2},\cdots ,x_{k-1}^{N_{p}}\right\}$ into $\left\{x_{k}^{1*},x_{k}^{2*},\cdots ,x_{k}^{N_{p}*}\right\}$ through the state equation. Each article in the latter cloud has weight 1/*N*_p_. When data y_k_ are available, the normalized weight of each particle is reevaluated, i.e., $W_{k}^{i}=\mathrm{Pr}\left(y_{k}|x_{k}^{i*}\right)/\sum_{j=1}^{N_{p}}\mathrm{Pr}\left(y_{k}|x_{k}^{j*}\right)$, where PDF $\mathrm{Pr}\left(y_{k}|x_{k}^{j*}\right)$ is defined by the measurement equation and knowledge of the statistical behavior of errors in data measurements. These weights are used for the estimation of posterior PDF Pr(**x***_k_*|**D***_k_*) [20]. 

Thus, once the posterior is available, the estimates of important statistics can be inferred. For instance, the minimum mean-squared error (MMSM) estimate is used in this paper, with
(34)$${\hat{x}}_{MMSE}\approx \frac{1}{N_{p}}\sum_{i=1}^{N_{p}}W_{i}x_{i}.$$

## 4. Simulation

The simulation was performed in a shallow water waveguide with a half-infinite elastic seabed as shown in Figure 1. The simulation environment parameters are shown in Table 1. The receiver was placed on the seafloor interface, and the range was 10 km. For convenience, *S*(*ω*) was constant.

Firstly, the effects of the particle size and the resampling strategies on the estimation performance were analyzed by simulation. Here, three widely used resampling algorithms (multinomial resampling, systematic resampling, and residual resampling) are discussed when the numbers of PF particles were 150, 300, 600, and 1200. The sound source depth was 20 m, the vertical wave impedance was the measured vector, and the frequency domain was selected as 50 Hz–150 Hz. The range estimation results and source depth estimation results in different resampling algorithms and particle sizes are shown in Table 2 and Table 3, respectively. It should be noted that each of the estimation values in Table 2 and Table 3 is the average of 20 simulation results. It can be seen from Table 2 and Table 3 that, as the number of particles increases, the positioning performances of the three resampling algorithms tend to increase. Following comprehensive comparison of the positioning results of the three resampling algorithms, the best one was determined as residual resampling. Thus, in all subsequent simulations and experiments, if not specified, the resampling algorithm selected residual resampling and the number of particles was chosen to be 1200.

Next, in the same simulation environment, the effect of **Q***_k_* (the covariance of state process noise) on the estimation performance was analyzed by simulation. As the initial value of **x** will affect the value of **Q***_k_*, in two different initial values, the influence of **Q***_k_* on the positioning performance was analyzed and the empirical rule for determining **Q***_k_* was obtained through simulation. The two different initial values of **x** were [9500, 25]^T^ and [7500, 25]^T^. 

When the initial value was [9500, 25]^T^, the different values of $Q_{k}^{1/2}$ are given in Table 4 and the localization results are shown in Figure 3. Combined with Figure 3 and Table 4, the following can be determined:

(i) Figure 3a,b show the positioning results when the second values on the diagonal of $Q_{k}^{1/2}$ were 0.5, 1, and 4, and the first value on the diagonal remained 100, which corresponds to cases (1a), (1b), and (1c), respectively. The distance and source depth cannot be correctly estimated in case (1a), but in cases (1b) and (1c), source position can be correctly estimated. At the same time, the source depth estimation curve in case 1(b) had a relatively smaller fluctuation around the true value compared to case 1(c). In all cases of (1a), (1b), and (1c), the positioning performance was best in case (1b);

(ii) Figure 3c,d show the positioning results when the first values on the diagonal of $Q_{k}^{1/2}$ were 10, 10^2^, and 10^3^, and the second value on the diagonal remained 1, which corresponds to cases (1d), (1b), and (1e), respectively. The distance and source depth cannot be correctly estimated in case (1d), but in cases (1b) and (1e), source position can be correctly estimated. Furthermore, the source depth estimation curve in case 1(e) had a relatively larger fluctuation around the true value compared to case 1(b). In all cases of (1d), (1b), and (1e), the positioning performance was best in case (1b);

(iii) The value of **Q***_k_* affects the positioning performance. Covariance matrix ***Q****_k_* should contain values large enough to accommodate the unexpected changes, but at the same time, the value of covariance matrix should not be too small which can cause poor positioning performance (such as in cases (1a) and (1d)). When the absolute difference between the initial value and true value of **x** was [500, 5]^T^, the positioning performance was best when $Q_{k}^{1/2}$ was $\left[\begin{array}{cc} 10^{2} & 0\\ 0 & 1 \end{array}\right]$ among the five cases.

When the initial value was [7500, 25]^T^, the different values of $Q_{k}^{1/2}$ are also given in Table 4 and the localization results are shown in Figure 4. Although values of $Q_{k}^{1/2}$ are different from that in the case of the initial value being [9500, 25]^T^, the empirical rule of selecting ***Q****_k_* obtained in Figure 4 is similar to that in Figure 3. The value of ***Q****_k_* relates to the absolute difference between the initial value and the true value of **x**, and the value of ***Q****_k_* cannot be selected too small, which will lead to failure in source localization as shown in the curves of (2a) and (2d) in Figure 4. Although the distance and source depth can be correctly estimated when the value of ***Q****_k_* is large enough, the estimated curves may have relatively larger fluctuations around the true value, for example, in the case of $Q_{k}^{1/2}$ being $\left[\begin{array}{cc} 2*10^{3} & 0\\ 0 & 1 \end{array}\right]$. In all cases of (2a)–(2e), the positioning performance was best when $Q_{k}^{1/2}$ was $\left[\begin{array}{cc} 10^{3} & 0\\ 0 & 1 \end{array}\right]$, i.e., case (2b). There was no fixed value for the covariance matrix ***Q****_k_*, but in the following simulations and experiments, the ***Q****_k_* value could be determined according to the empirical rule obtained from these simulations.

Then, in the same simulation conditions, the estimation performances of EKF and the unscented Kalman filter (UKF) were compared with PF. The localization results are shown in Figure 5. It can be seen from Figure 5a that UKF and PF performed better than EKF in terms of range estimation. However, in terms of source depth estimation, both UKF and EKF estimation curves had larger fluctuations near the true value (20 m) than the PF estimation curve, as shown in Figure 5b. Figure 5 shows that the positioning performance of PF is superior to both EKF and UKF.

Then, under different source depths conditions, we discussed the source localization performance of pressure and vertical wave impedance as the measurement vector.

### 4.1. Source Depth of 20 m

When the sound source depth was 20 m, the measured vectors were the pressure and vertical wave impedance. The frequency domain was selected as 25 Hz–100 Hz, and the localization results are shown in Figure 6. For both pressure and vertical wave impedance, the range estimation results converged and were in good agreement with the true range of 10 km (Figure 6a,c). As can be seen from Figure 6b and Figure 6d, both depth iteration curves converged to the true value of 20 m; however, compared to Figure 6d, the curve in Figure 6b (the pressure as the measurement vector) had large fluctuation near the true source depth. In Table 5, the mean absolute percentage error (MAPE) values of localization results are given for comparing the positioning performance between pressure and vertical wave impedance. MAPE was calculated as the average of the unsigned percentage error, as shown in the example below.
(35)$$\mathrm{MAPE}=\left(\frac{1}{N}\sum_{n=1}^{N}\frac{\left|{\hat{x}}_{n}-x_{true}\right|}{\left|x_{true}\right|}\right)*100,$$where *N* is the number of estimates. Here, the estimate values were selected from iterations 20 to 100. It can be clearly seen from Table 5 that the pressure was slightly better than the vertical wave impedance when estimating the distance, but neither MAPE exceeded 0.02%. Conversely, the vertical wave impedance was better in estimating the source depth than pressure.

Next, different frequency bands (50 Hz–150 Hz) were selected for the simulation, and the localization results are shown in Figure 7. The MAPE values of localization results are given in Table 6. At 50 Hz–150 Hz, the positioning performances of pressure and vertical wave impedance were basically the same as those at 25 Hz–100 Hz except that the MAPE values were slightly different. Under the simulation conditions, Figure 6 and Figure 7, and Table 5 and Table 6 prove that the positioning performances of pressure and vertical wave impedance were not affected by the frequency band too much. 

### 4.2. Source Depth of 40 m

When the sound source depth was 40m, the measured vectors were the pressure and vertical wave impedance. The frequency domain was selected as 75 Hz–95 Hz, and the localization results are shown in Figure 8. In this case, the performance of vertical wave impedance for localization was still good, as shown in Figure 8c,d. However, when the pressure was used for distance estimation, the result was extremely poor, as shown in Figure 8a, where the iteration estimation curve had great fluctuation at the true value of 10 km. Figure 8b shows that the source depth estimation results converged to the true value of 40 m, but the curve fluctuated within 35 m–45 m. Upon increasing the number of iterations and simulating in the same environment, the localization results of the pressure are shown in Figure 9. Increasing the number of iterations still could not change the ranging capability of the pressure.

The reason for the appearance of Figure 8 is given from the perspective of sensitivity analysis. The effect of source localization depends on the sensitivity of parameters. The sensitivity of parameters refers to the change degree of the objective function caused by the change of the parameter. The normalized objective function used here is given by
(36)$$\Phi \left(x\right)=1-\frac{1}{y^{H}y}{\left\|y-\frac{d{\left(x\right)}^{H}yd\left(x\right)}{{\left\|d\left(x\right)\right\|}^{2}}\right\|}^{2},$$where **y** is the actual measurement vector. H represents the conjugate transpose. When the pressure is used as the measurement vector, that is, **y** = [*p*(**x**, *ω*_1_), …, *p*(**x**, *ω_m_*), …, *p*(**x**, *ω*_M_)]^T^, and *p*(**x**, *ω_m_*) is the pressure at angular frequency *ω_m_*, with m∈[1,M]. Furthermore, **d**(**x**) = [d_1_(**x**), …, d*_m_*(**x**), …, d_M_(**x**)]^T^, with $d_{m}\left(x\right)=2\pi iS\left(\omega_{m}\right)\rho_{1}\omega_{m}^{2}\sum_{n}F_{nm}^{2}\sin\beta_{1nm}H\sin\beta_{1nm}z_{s}H_{0}^{\left(1\right)}\left(\xi_{nm}r\right)$. When the vertical wave impedance is used as the measurement vector, that is, **y** = [*Z_z_*(**x**, *ω*_1_), …, *Z_z_* (**x**, *ω_m_*), …, *Z_z_* (**x**, *ω*_M_)]^T^, and *Z_z_* (**x**, *ω_m_*) is the vertical wave impedance. Now, **d**(**x**) = [d_1_(**x**), …, d*_m_*(**x**), …, d_M_(**x**)]^T^, with $d_{m}\left(x\right)=i\rho_{1}\omega_{m}\frac{\sum_{n}F_{nm}^{2}\sin\beta_{1nm}H\sin\beta_{1nm}z_{s}H_{0}^{\left(1\right)}\left(\xi_{nm}r\right)}{\sum_{n}F_{nm}^{2}\beta_{1nm}\cos\beta_{1nm}H\sin\beta_{1nm}z_{s}H_{0}^{\left(1\right)}\left(\xi_{nm}r\right)}$. When the measured vectors were the pressure and vertical wave impedance in the frequency domain of 75 Hz–95 Hz, the objective functions for the state vector **x** are given in Figure 10. When the measured vector was the vertical wave impedance, the range and source depth were very sensitive to the objective function, and the objective functions reached the maximum at 10 km and 40 m. When the measured vector was the pressure, the source depth was sensitive to the objective function. This is why depth estimation could be performed using pressure. However, the range was not sensitive to the objective function, whereby the objective function curve is represented by the black line in Figure 10a. The curve peaked at multiple distances, such as 6 km, 10 km, etc. This multi-peak phenomenon led to instability of ranging performance, as shown in Figure 8a and Figure 9a.

Above all, the source depth will affect the correctness of localization when pressure is used as the measurement vector, while the use of vertical wave impedance to locate the source is not affected by the source depth. In addition, there is no need to know the source spectrum information when using vertical wave impedance for positioning, which is an advantage for passive ranging. In Section 5, the vertical wave impedance is used to locate the explosive sources during an experiment based on the PF.

## 5. Experimental Results

In 2018, an underwater explosion experiment was conducted in a shallow water waveguide near Qingdao, China. The experimental ship sailed along the scheduled route, and the explosives were placed at a fixed point during the voyage. Two vertical arrays were used as receiving devices. Each vertical array consisted of two hydrophones and one OBS. The navigation trajectory of the experimental ship and the coordinates of the receiving array were obtained from the global positioning system (GPS) data, as shown in Figure 11.

During the experiment, the temperature was recorded by a temperature and depth logger (TD) with a tiny change. Due to the little temperature variability and the small depth of the water column, water sound speed was assumed to be constant. The bottom of the sea could be assumed to be flat, which was estimated as a semi-infinite uniform elastic seabed.

During the experiment, a total of 19 explosives were detonated at four burst spots, and the third and fourth burst spots were selected for sound source localization. Seven explosives (No. 9–No. 15) were selected for source localization. For the seven explosives, the explosive quantity was 50 g. At the third burst spot, four explosives (No. 9–No. 12) were carried out with a depth of 13 m. At the fourth burst spot, three explosives (No. 13–No. 15) were carried out with a depth of 17 m.

Because the second vertical array was on the slope and did not conform to the environment model in this paper, only the data received by the OBS of the first array were used for positioning. 

The PF positioning results of experimental data received by the OBS for No. 9–No. 15 explosions are illustrated in Figure 12. The black solid line denotes the estimated value, and the red dashed line denotes the actual distance obtained by the GPS data, with the actual distances shown in Table 7. Moreover, Table 7 also gives the MAPE of localization results for No. 9–No. 15 explosions. In the process of using the PF for experimental data, the state process noise covariance value was taken according to the empirical rule obtained in the Section 4. Here, the initial state value was [21000, 15]^T^ and the $Q^{1/2}$ was $\left[\begin{array}{cc} 10^{2} & 0\\ 0 & 2 \end{array}\right]$ for No. 9–No. 12 explosions. Additionally, the initial state value was [29000, 15]^T^ and the $Q^{1/2}$ was $\left[\begin{array}{cc} 10^{2} & 0\\ 0 & 2 \end{array}\right]$ for No. 13–No. 15 explosions. Figure 12 indicates that the experimental range estimates and source depth estimates both converged, and that the differences between the convergence values and the true values were small for No. 9–No. 15 explosions. The MAPEs of range estimation and depth estimation for No. 9–No. 12 did not exceed 1.42% and 2.86%, respectively, while the MAPEs of range estimation and depth estimation for No. 13–No. 15 did not exceed 0.29% and 3.30%. The experimental results proved the accuracy and practicability of the method using vertical wave impedance as a measurement vector based on the PF.

## 6. Conclusions

In this paper, an OBS sensor was used to estimate the position of the broadband sound source in a Pekeris shallow water waveguide with elastic bottom. In the semi-infinite elastic seabed environment, the expression of pressure and vertical velocity channels received by the OBS sensor were theoretically derived. Based on the PF method, the positioning performances of the pressure and vertical wave impedance as the measurement vector were simulated and analyzed. In the simulation environment, the results showed that the source depth will affect the ability of pressure to locate. When the measured vector was the pressure and the source was 40 m, although the source depth was sensitive to the objective function, the range was not sensitive to the objective function. Moreover, the objective function curve of range peaked at multiple distances. This multi-peak phenomenon led to instability of ranging performance. In contrast, when the measured vector was the vertical wave impedance and the source was 20 m, the range and source depth estimation results converged and were in good agreement with the true values at different frequency bands. In the case of poor localization performance for pressure, that is, when the source was 40 m, the vertical wave impedance as the measurement vector could also exhibit excellent positioning performance. The PF method with the vertical wave impedance as the measurement vector was not affected by source depth and source spectrum information, making it more tolerant and more robust than that with pressure in positioning.

The PF method with the vertical wave impedance as the measurement vector was employed to process the experimental data in the sea near Qingdao. The experimental results showed that the source parameters (the source depth and the range) were correctly estimated and converged to the true values using an OBS sensor in different ranges and different explosive source depths.

## Figures and Tables

**Figure 1 sensors-19-02236-f001:**
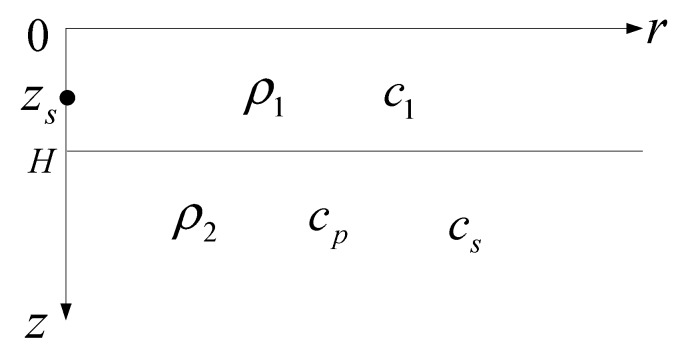
The Pekeris waveguide with elastic bottom.

**Figure 2 sensors-19-02236-f002:**
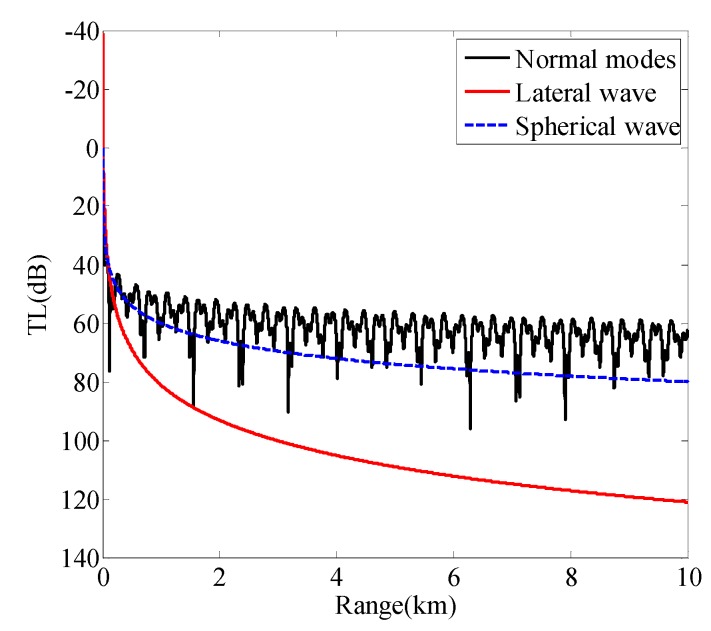
Transmission losses (TLs) of normal modes (black line), the lateral wave (red line), and the spherical wave (blue dashed line).

**Figure 3 sensors-19-02236-f003:**
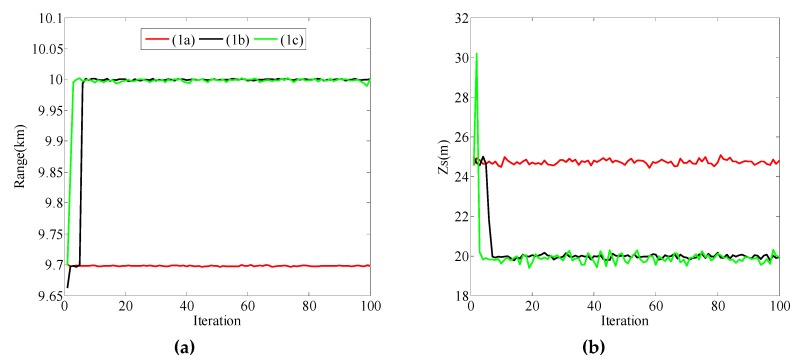
Localization results for particle filtering (PF) with different values of $Q_{k}^{1/2}$ when the initial value of **x** was [9500, 25]^T^. (**a**) Range estimation results in cases (1a), (1b), and (1c); (**b**) source depth estimation results in cases (1a), (1b), and (1c); (**c**) range estimation results in cases (1d), (1b), and (1e); (**d**) source depth estimation results in cases (1d), (1b), and (1e).

**Figure 4 sensors-19-02236-f004:**
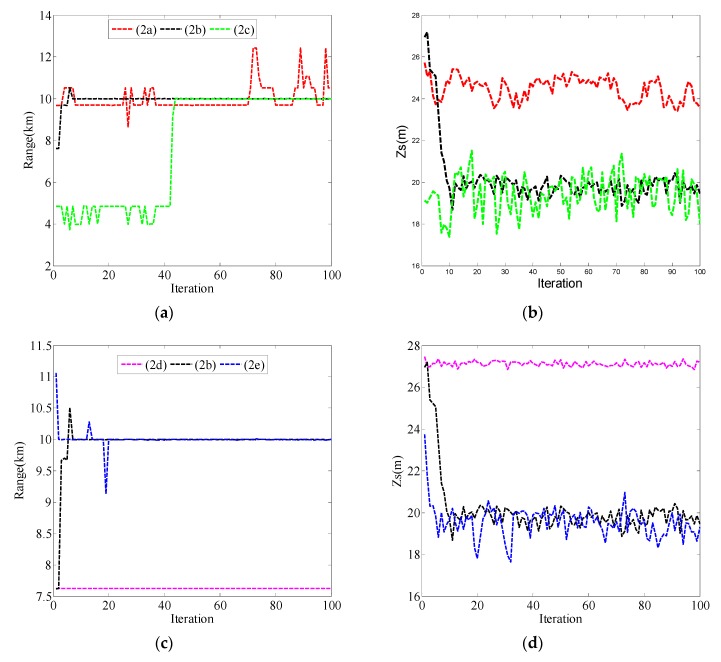
Localization results for PF with different values of $Q_{k}^{1/2}$ when the initial value of **x** was [7500, 25]^T^. (**a**) Range estimation results in cases (2a), (2b), and (2c); (**b**) source depth estimation results in cases (2a), (2b), and (2c); (**c**) range estimation results in cases (2d), (2b), and (2e); (**d**) source depth estimation results in cases (2d), (2b), and (2e).

**Figure 5 sensors-19-02236-f005:**
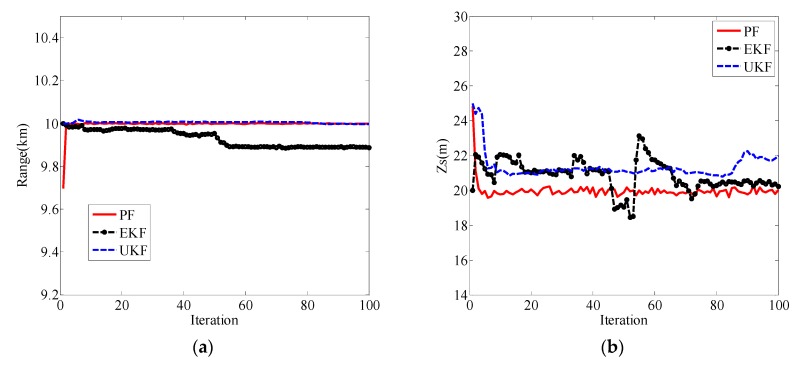
Localization results for PF, extended Kalman filter (EKF), and unscented Kalman filter (UKF). (**a**) Range estimation results for PF, EKF, and UKF; (**b**) source depth estimation results for PF, EKF, and UKF.

**Figure 6 sensors-19-02236-f006:**
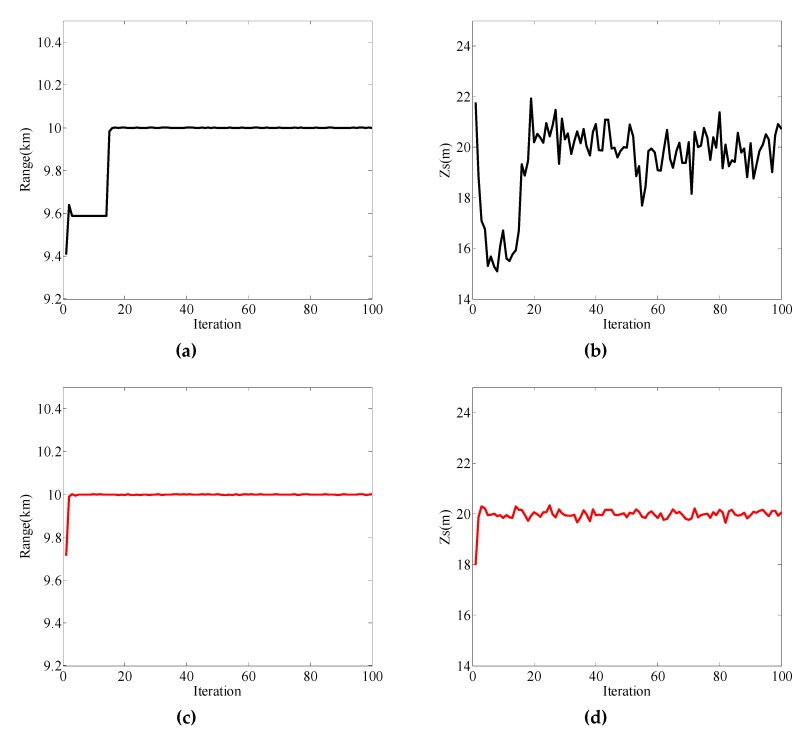
Localization results for the measured vector as the pressure (black line) and the vertical wave impedance (red line). (**a**) Range estimation result with pressure as the measured vector; (**b**) source depth estimation result with pressure as the measured vector; (**c**) range estimation result with vertical wave impedance as the measured vector; (**d**) source depth estimation result with vertical wave impedance as the measured vector.

**Figure 7 sensors-19-02236-f007:**
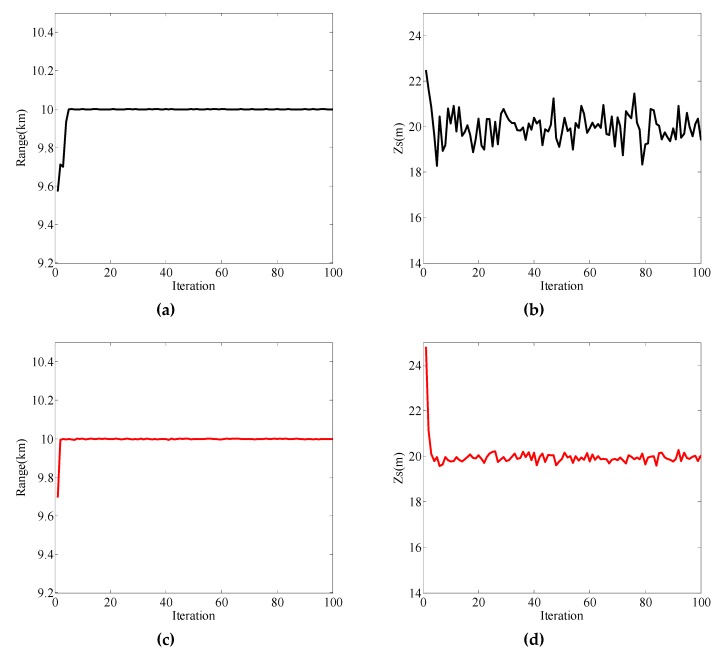
Localization results for the measured vector as the pressure (black line) and the vertical wave impedance (red line). (**a**) Range estimation result with pressure as the measured vector; (**b**) source depth estimation result with pressure as the measured vector; (**c**) range estimation result with vertical wave impedance as the measured vector; (**d**) source depth estimation result with vertical wave impedance as the measured.

**Figure 8 sensors-19-02236-f008:**
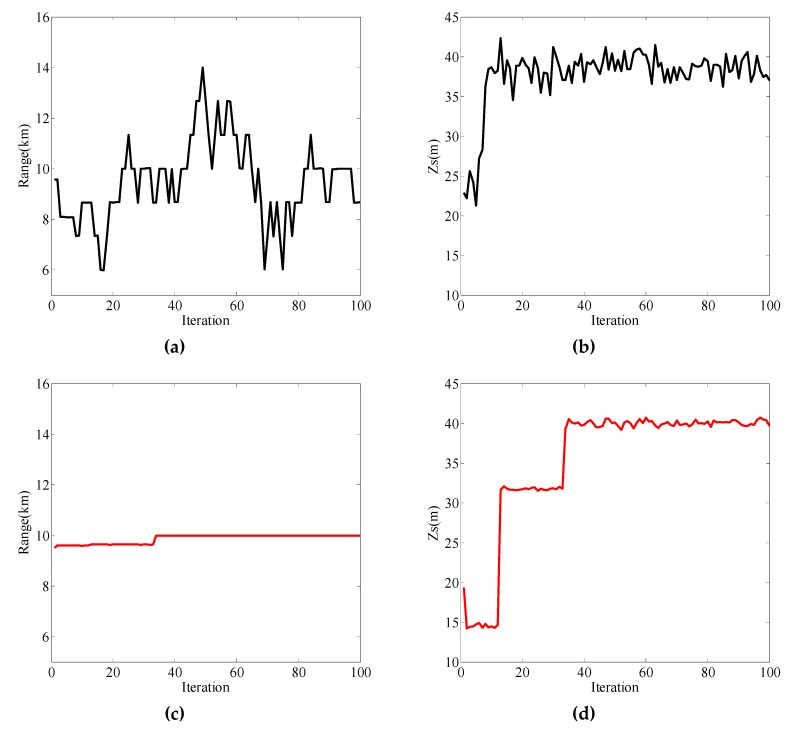
Localization results for the measured vector as the pressure (black line) and the vertical wave impedance (red line). (**a**) Range estimation result with pressure as the measured vector; (**b**) source depth estimation result with pressure as the measured vector; (**c**) range estimation result with vertical wave impedance as the measured vector; (**d**) source depth estimation result with vertical wave impedance as the measured vector.

**Figure 9 sensors-19-02236-f009:**
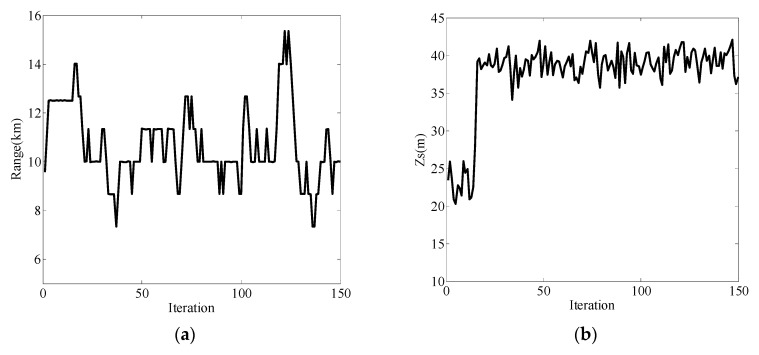
Localization results for pressure as the measured vector. (**a**) Range estimation result with pressure as the measured vector; (**b**) source depth estimation result with pressure as the measured vector.

**Figure 10 sensors-19-02236-f010:**
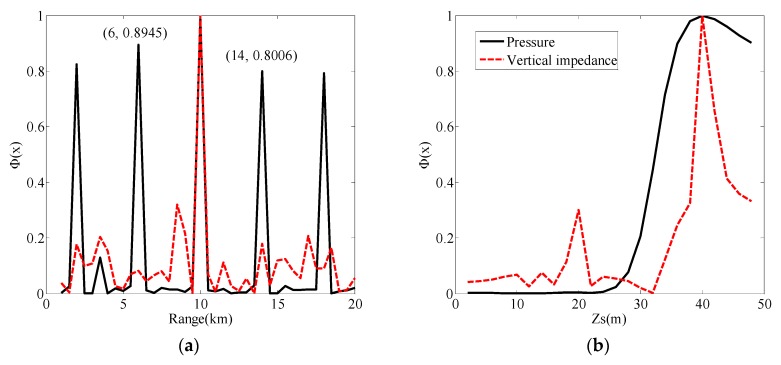
Normalized objective function curves. (**a**) Normalized objective functions for the range when the measured vector was the pressure (black line) and the vertical wave impedance (red dotted line); (**b**) normalized objective functions for source depth when the measured vector was the pressure (black line) and the vertical wave impedance (red dotted line).

**Figure 11 sensors-19-02236-f011:**
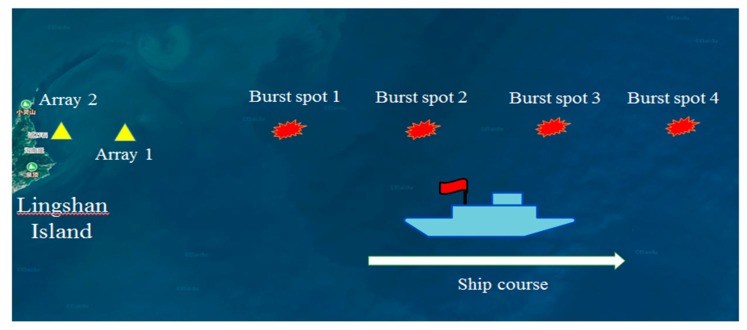
Navigation trajectory of the experimental ship and localization of the receiving arrays.

**Figure 12 sensors-19-02236-f012:**
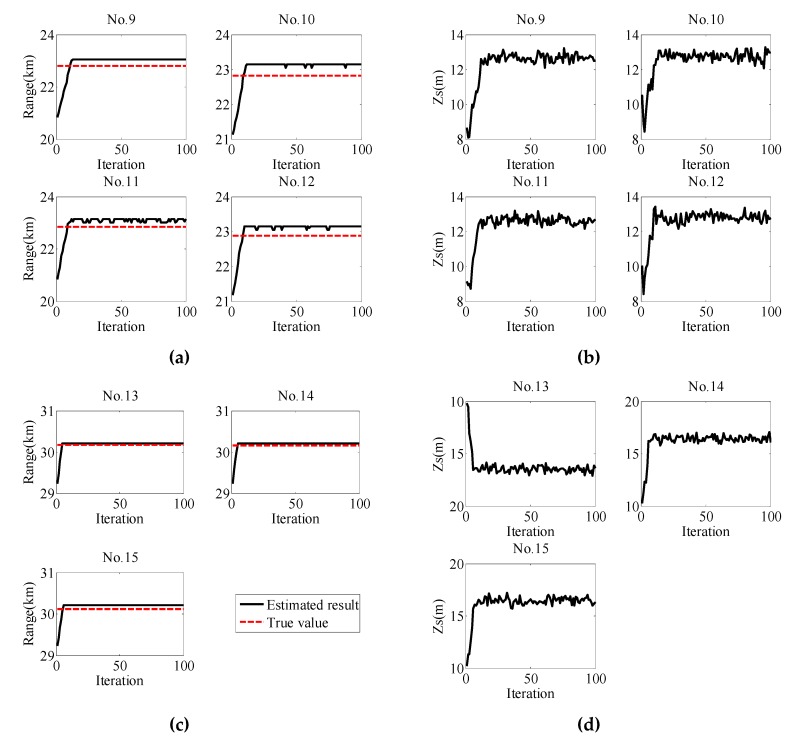
PF positioning results for the No.9–No.15 explosions. (**a**) Range estimation results for the No.9–No.12 explosions; the estimated results and the global positioning system (GPS) ranges are represented by the black line and red dotted line, respectively; (**b**) source depth estimation results for the No.9–No.12 explosions; (**c**) range estimation results for the No.13–No.15 explosions; the estimated results and the GPS ranges are represented by the black line and red dotted line, respectively; (**d**) source depth estimation results for the No.13–No.15 explosions.

**Table 1 sensors-19-02236-t001:** Parameters of the ocean environment.

Medium	Depth(m)	Density(g/cm^3^)	Compression Wave Speed (m/s)	Shear Wave Speed (m/s)
Fluid	50	1.0	1500	/
Elastic bottom	/	1.5	3800	1800

**Table 2 sensors-19-02236-t002:** Range estimation results in different resampling algorithms and particle sizes.

Particle Size	Multinomial Resampling	Systematic Resampling	Residual Resampling
150	9904.4 m	9869.6 m	9902.5 m
300	9904.0 m	9910.4 m	9894.5 m
600	9957.9 m	9907.1 m	9941.0 m
1200	9938.0 m	9949.1 m	9950.3 m

**Table 3 sensors-19-02236-t003:** Source depth estimation results in different resampling algorithms and particle sizes.

Particle Size	Multinomial Resampling	Systematic Resampling	Residual Resampling
150	25.6 m	28.8 m	25.4 m
300	25.7 m	25.2 m	26.3 m
600	22.6 m	26.3 m	23.7 m
1200	22.6 m	22.1 m	21.1 m

**Table 4 sensors-19-02236-t004:** Different values of $Q_{k}^{1/2}$ in different initial values of **x**.

**The Initial Value of** **x** **Was [9500, 25]^T^**					
**No.**	**(1a)**	**(1b)**	**(1c)**	**(1d)**	**(1e)**
$Q_{k}^{1/2}$	$\left[\begin{array}{cc} 10^{2} & 0\\ 0 & 0.5 \end{array}\right]$	$\left[\begin{array}{cc} 10^{2} & 0\\ 0 & 1 \end{array}\right]$	$\left[\begin{array}{cc} 10^{2} & 0\\ 0 & 4 \end{array}\right]$	$\left[\begin{array}{cc} 10 & 0\\ 0 & 1 \end{array}\right]$	$\left[\begin{array}{cc} 10^{3} & 0\\ 0 & 1 \end{array}\right]$
**The Initial Value of x Was [7500, 25]^T^**					
**No.**	**(2a)**	**(2b)**	**(2c)**	**(2d)**	**(2e)**
$Q_{k}^{1/2}$	$\left[\begin{array}{cc} 10^{3} & 0\\ 0 & 0.5 \end{array}\right]$	$\left[\begin{array}{cc} 10^{3} & 0\\ 0 & 1 \end{array}\right]$	$\left[\begin{array}{cc} 10^{3} & 0\\ 0 & 4 \end{array}\right]$	$\left[\begin{array}{cc} 10^{2} & 0\\ 0 & 1 \end{array}\right]$	$\left[\begin{array}{cc} 2*10^{3} & 0\\ 0 & 1 \end{array}\right]$

**Table 5 sensors-19-02236-t005:** Mean absolute percentage error (MAPE) of localization results in the frequency domain of 25 Hz–100 Hz.

	Range	Source Depth
Pressure	0.0023%	2.7709%
Vertical wave impedance	0.0107%	0.5129%

**Table 6 sensors-19-02236-t006:** MAPE of localization results in the frequency domain of 50 Hz–150 Hz.

	Range	Source Depth
Pressure	0.0031%	2.2607%
Vertical wave impedance	0.0117%	0.6860%

**Table 7 sensors-19-02236-t007:** Distances of the explosives from the first array and the MAPEs of the localization results.

No.	9	10	11	12	13	14	15
Distance (m)	22,801	22,822	22,844	22,882	30,176	30,158	30,122
MAPE (%) for range	1.11	1.42	1.14	1.14	0.11	0.17	0.29
MAPE (%) for source depth	2.55	2.13	2.86	1.92	3.10	3.30	3.21

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
