# Peer review of "Particle Filtering for Localization of Broadband Sound Source Using an Ocean-Bottom Seismometer Sensor"

_sensors, 2019, doi:10.3390/s19102236_

Reviewer 1 Report

The paper presents the broadband sound source localisation using the particle filtering and the method is further validated using experimental data in seas.

The paper presents an application-oriented research, which employs the standard PF, leaving almost no contribution in algorithm or theoretical aspects. However, as respect to the PF implementation, the authors still have the following concerns:

1. the system dynamic (24) is essentially static as x(k)=x(k-1)+w(k-1). This makes the application of PF problematic, instead the MCMC methods should be more relevant. Furthermore, how to determine the statistical property of w(k-1), such as covariance in practice?

2. To justify the superiority of the PF in the proposed sound source localisation, it should be compared with the performance of other nonlinear estimation algorithms, especially the standard ones such as EKF, UKF. 

3. More details and analysis of the PF implementation in the simulation & experimental sections should be provided, such as the effect of the particle size, the re-sampling strategies on the estimation performance.

Author Response

Dear Reviewer:

Thank you very much for giving us an opportunity to revise our manuscript.

The responses to your comments can be found in the attachment (a word file).

Thank you for your attention.

From the authors

Reviewer 2 Report

The paper titled “Particle filtering for localization of broadband sound source using an Ocean Bottom seismometer sensors” by Liu et al. presents a novel technique to calculate a source using vertical velocity and pressure measure at the sea bottom under the Pekeris environment with elastic bottom. The technique also apply to real data as well as synthetic ones. The results is reasonable to support authors claims. However, a part of methods is wordy. In addition, some of figures are intentionally shown with using differential scale of axis. The results, however, are interesting, and the paper is well written. I believe that its results are highly relevant as per the scope of Sensor. I recommend the Editor to accept the paper after the following some issues are resolved.

Minor issues:

1.     The authors have thoroughly mentioned their method proposed in this paper, which has been started from a theoretical background of acoustic wave field under the Pekeris environment in the cylindrical coordinate system. However, the most of the portions, especially in 2.1 has been well documented. I strongly recommend to move the part to supplemental material or an appendix.

2.     Lines 133-134, authors proposed that the contribution of lateral wave in equation (19) can be neglected under the condition of long range sound transmission. However, the equation 19 probably shows the same dependency both of normal mode and lateral wave on transmitted distance. I ask authors to show reason why the lateral wave, or lateral component can be neglect explicitly. This explanation will be also important for reader because authors is using only normal mode with several kilometer distance, which will be not long distance enough to be ignored.

3.     In Figures 2, 3, and 4,  authors use different scales to plot the results. I strongly recommend to use the same scale in each figure at least, to compare the performances of two methods using pressure and vertical impedance as data.

Author Response

(The authors gave the same response as above.)

Reviewer 3 Report

Please review the proposal for minor fixes, for instance:

25-26:
"...wave impedance for localization the explosive source." =>
"...wave impedance for the localization of the explosive source."
30:
"Estimating the location of an underwater sound source.." =>
"Localization of an underwater sound source.."
49-51:
"Besides, successful localization with real data was demonstrated using arrival times and corresponding probability density functions (pdfs) extracted via particle filtering."
Provide a reference.
55:
"liquid seabed" => "fluid seabed"
56:
"...the effects of shear wave." => "...the effects of shear waves."
56-57:
"...is generally elastic medium." => "...is generally an elastic medium."
58:
"...Pekeris waveguide with elastic bottom." => "...Pekeris waveguide with an elastic bottom."
80:
"The solid bottom..." => "The elastic bottom..."
99:
"...are zero order and first order Bessel function..." =>
"...are zero order and first order Bessel functions..."
112:
"...tangential stress equals to zero." =>
"...tangential stress equal to zero."
151: use parenthesis in the expression for the impedance
159-160:
"...applied to nonlinear system..." =>
"...applied to nonlinear systems..."
281:282:
" ...S(w) takes a constant that does not change with w." =>
" ...S(w) is constant." 
Figure8: "Ture" => "True"

Author Response

Dear Reviewer:

Thank you very much for giving us an opportunity to revise our manuscript.

The responses to your comments can be found in the attachment (a word file).

Thank you for your attention.

From the authors

Round  2

Reviewer 1 Report

The reviewer is satisfied with the revision except the reply to comment 1. Please elaborate how to determine the covariance through simulation and furthermore how to get around it in real-world application.

Author Response

(The authors gave the same response as above.)
